# Secrecy Capacity Maximization of UAV-Enabled Relaying Systems with 3D Trajectory Design and Resource Allocation

**DOI:** 10.3390/s22124519

**Published:** 2022-06-15

**Authors:** Qi An, Yu Pan, Huizhu Han, Hang Hu

**Affiliations:** School of Information and Navigation, Air Force Engineering University, Xi’an 710077, China; panyu@mail.nwpu.edu.cn (Y.P.); h18137692630@163.com (H.H.); xd_huhang@126.com (H.H.)

**Keywords:** unmanned aerial vehicle (UAV) communication, mobile relay, secrecy capacity, joint optimization algorithm

## Abstract

Unmanned aerial vehicles (UAVs) have attracted considerable attention, thanks to their high flexibility, on-demand deployment and the freedom in trajectory design. The communication channel quality can be effectively improved by using UAV to build a line-of-sight communication link between the transmitter and the receiver. Furthermore, there is increasing demand for communication security improvement, as the openness of a wireless channel brings serious threat. This paper formulates a secrecy capacity optimization problem of a UAV-enabled relay communication system in the presence of malicious eavesdroppers, in which the secrecy capacity is maximized by jointly optimizing the UAV relay’s location, power allocation, and bandwidth allocation under the communication quality and information causality constraints. A successive convex approximation–alternative iterative optimization (SCA-AIO) algorithm is proposed to solve this highly coupled nonconvex problem. Simulation results demonstrate the superiority of the proposed secrecy transmission strategy with optimal trajectory design and resource allocation compared with the benchmark schemes and reveal the impacts of communication resources on system performance.

## 1. Introduction

### 1.1. Background and Related Works

Compared with the traditional terrestrial communication, the unmanned aerial vehicle (UAV) is regarded as an important component in emergency communication, mobile internet, and Internet of Things (IoT) due to its rapid response, strong flexibility and low cost [1,2]. In recent years, UAVs have been used in many communication scenarios such as aerial base station (BS) [3], mobile relay [4], flying cellular user [5], communication blind compensation [6], and airborne edge-cloud computing [7]. As one of the main applications of a UAV communication system, the UAV relay technology can effectively solve the wireless traffic congestion as well as the dynamic and unbalanced distribution of mobile devices [8,9,10].

The flexibility to build a line-of-sight (LoS) link is one of the main advantages of UAV-enabled communication [11]. However, the LoS channel is also vulnerable to be eavesdropping by malicious users [12,13,14] due to the channel characteristics of opening and broadcasting. Meanwhile, wireless channels have the characteristics of opening and broadcasting. Therefore, the security of UAV-enabled communication systems needs to be taken seriously, which was not considered in [3,4,5,6,7,8,9,10]. Due to the limited resources and strong flexibility of the UAV, the traditional transmission method could not be directly used in the UAV-enabled communication. As an important supplement of encryption technologies, physical-layer security (PLS) realizes the secrecy transmission by increasing the channel capacity of legitimate users and reducing the channel capacity of eavesdropping users [15,16,17,18,19,20,21].

Trajectory design and resource allocation have been typical technique optimization ways to enhance security for UAV-enabled communication systems. A joint bandwidth allocation and UAV two-dimensional (2D) trajectory design with fixed altitude was proposed in [15] to maximize the secrecy rate of a frequency division duplex (FDD)-based UAV relay system. Ref. [16] optimized the trajectory and transmit power allocation of a UAV BS to maximize the average worst-case secrecy rate. To enchance the security of the UAV-enabled relay system, the authors in [17] proposed a deep reinforcement learning algorithm to optimize the 2D trajectory and power allocation of the UAV. A dual UAV-assisted communication system was considered in [18], where the two UAVs are used as a base station and friendly jammer, respectively, and then, the average secrecy rate was maximized by optimizing the trajectories and power allocation of the two UAVs.

On the other hand, artificial noise (AN) and beamforming are popular techniques that can further guarantee security of the UAV-enabled communication. Ref. [19] proposed a secrecy two-phase transmission protocol via AN-assisted UAV communications and enhanced the average secrecy rate by jointly optimizing the UAV’s trajectory, transmission power, and AN power. Ref. [20] considered a UAV swarm-enabled communication system, in which the PLS of the system is improved by beamforming and bandwidth allocation in the presence of an eavesdropper. The authors in [21] studied a UAV swarm-enabled communication network, where several rotary-UAVs act as mobile relays to serve a legitimate UAV user in the presence of an eavesdropper UAV. Then, the power allocation coefficient, beamforming on UAV relays, and trajectory of the legitimate UAV are jointly optimizated to maximize the secrecy rate.

### 1.2. Motivation and Contributions

As mentioned above, the existing UAV trajectory optimization methods generally focus on 2D optimization under fixed flight altitude, which can not take full advantage of UAV’s maneuverability. Furthermore, most studies of PLS in UAV-enabled communication rely on static ground users without considering the existence of mobile users in many scenarios. There is still a lack of research contribution on the effect of mobile ground users on the UAV communication systems, and the corresponding secrecy capacity optimization scheme has not been studied yet. Meanwhile, mobile ground users may bring more challenges in practice, such as how to maintain reliable LoS communication and satisfy the quality of service (QoS) requirements of mobile users.

Based on the above description and discussion, this paper mainly investigates the 3D trajectory optimization and resource allocation to improve the secrecy performance of the UAV-enabled relaying system, where the reliability of LoS communication and QoS of mobile ground users are considered. In particular, a rotary UAV is used as a relay to assist the communication from the BS to multiple mobile users which are monitored by a malicious user; then, the power allocation, bandwidth allocation, and trajectory of the UAV are jointly optimized to maximize the secrecy channel capacity. To handle the nonconvex optimization problem, the SCA-AIO algorithm is utilized. The contributions of this article can be summarized as follows:A UAV-enabled relaying communication system in the presence of an eavesdropper is proposed in this paper. To enhance the secrecy capacity of this system, we formulate a problem by jointly optimizing the UAV’s 3D trajectory and resource allocation under the constraints on information-causality, LoS communication reliability, and QoS requirements of mobile users.For the highly coupled nonconvex problem, we develop an efficient algorithm to solve it. Specifically, we decompose the optimization problem into three subproblems and transform each nonconvex subproblem into a solvable convex optimization problem by the SCA algorithm. Among them, for the power and bandwidth allocation problems, we transform all the constraints into second-order cone (SOC) constraints and linear matrix inequality (LMI) constraints by introducing auxiliary variables to improve the solution efficiency. Meanwhile, we propose a joint optimization algorithm based on the AIO algorithm framework and analyze the convergence and complexity of the algorithm.The following fruitful insights are obtained: (1) Numerical results show that the proposed scheme has better performance in terms of the secrecy capacity compared with the traditional optimization schemes. (2) Based on the theoretical analysis and simulation results, the path panning of the UAV relay-adjusted coordinate according to the position of the mobile users is analyzed qualitatively. (3) According to in-depth simulations, we provide the impacts of different parameter configurations on the performance of a UAV-enabled relaying system.

### 1.3. Organization

The rest of this paper is organized as follows. Section 2 presents the system model and states the problem formulation. In Section 3, we propose a joint optimization algorithm to solve the secrecy capacity maximization problem and analyze the complexity as well as the convergence of the proposed algorithm. Simulation results are provided in Section 4. Finally, conclusions are drawn in Section 5.

## 2. System Model and Problem Formulation

As shown in Figure 1, we consider a UAV communication system in which multiple mobile users need to receive information from the BS, while an eavesdropper EVE is located around mobile users and attempts to intercept the downlink signal between UAV and mobile users. There is no direct communication link between BS and the users due to obstacle occlusion. Therefore, we establish a two-hop wireless relaying transmission link by using a UAV relay. Without loss of generality, a three-dimensional (3D) Cartesian coordinate system is considered. Suppose that a set of Ω={1,2,3,…,N} mobile users are randomly distributed on the ground, and the UAV, which is denoted as R, can adjust its 3D position. Then, the coordinates of BS, UAV, and the *n*th mobile user are $q_{BS}=\left(0,0,0\right)$, $q_{R}=\left(x_{R},y_{R},z_{R}\right)$, and $q_{n}=\left(x_{n},y_{n},0\right)$, respectively. The rough location of EVE can be detected by an optical camera or synthetic aperture radar (SAR) [22]. In practice, the eavesdroppers always try to hide their existence, and the location detection may suffer from errors [23]. Therefore, we assume that the error between the measured value and the actual coordinates is $r_{mk}$.

According to [17], the probability of LoS communication between the UAV R and the ground communication node m is
(1)$$P_{{\mathrm{LoS}}_{R,\;m}}=\frac{1}{1+\varphi \exp\left(-\xi \left[\theta_{R,m}-\varphi \right]\right)},m\in \left\{\mathrm{BS},\Omega \right\}$$
where φ and ξ are constants depending on the environment type, while $d_{m,R}=∥q_{m}-q_{R}∥$ and $\theta_{m,R}=\frac{180}{\pi }\sin^{-1}\left(\frac{z_{R}}{d_{m,R}}\right)$ are the distance and elevation angle between the UAV and node m, respectively.

To ensure the reliability of the LoS communication between UAV and the BS/mobile users, the probability of LoS needs to meet $P_{\mathrm{LoS}}\geq \varepsilon_{\mathrm{LoS}}$ [24], where the $\varepsilon_{\mathrm{LoS}}$ is a constant close to 1. To facilitate the analysis, this constraint can be rewritten as follows
(2)$$\sin\left(\varphi +\frac{1}{\xi }\ln\frac{\varphi \varepsilon_{\mathrm{LoS}}}{1-\varepsilon_{\mathrm{LoS}}}\right)\cdot d_{m,R}\leq z_{R}$$

To guarantee the security of information transmission, we consider the worst case, i.e., it is a LoS link between the UAV and EVE. The Doppler effect is assumed to be perfectly compensated [25]. Then, the channel gain between the UAV and BS, EVE, and the *n*th mobile user can be expressed as the following free-space path loss model [26],
(3)$$h_{X,R}=\beta_{0}{\left(d_{X,R}\right)}^{-2}$$
where X∈{BS,EVE,Ω}, and $\beta_{0}$ denotes the channel power gain at the reference distance *d* = 1 m, which depends on the carrier frequency, antenna gain, etc., and $d_{X,R}$ is the distance between X and R. Considering the worst case, the minimum distance between the UAV R and the EVE is $d_{EVE,R}=\sqrt{{\left(\sqrt{{\left(x_{R}-S\right)}^{2}+{\left(y_{R}-D\right)}^{2}}-r_{mk}\right)}^{2}+z_{R}^{2}}$.

We assume that the downlink between UAV and mobile users adopts frequency division multiple access (FDMA), and the UAV adopts the decode-and-forward (DF) relaying method. The UAV transmits information to mobile users with bandwidth allocation $B=\left\{B_{n,R},\forall n\in \Omega \right\}$ and power allocation $P=\left\{P_{n,R},\forall n\in \Omega \right\}$. Considering the limited spectrum resources, the maximum bandwidth for this downlink is denoted as $B_{\max}$, and the maximum transmitting power of the UAV is $P_{\max}$. The channel capacity between the UAV and the *n*th mobile user is
(4)$$C_{n}=B_{n,R}r_{n,R}$$
where $r_{n,R}=\log_{2}\left(1+\frac{P_{n,R}\delta_{n}}{d_{n,R}^{2}}\right)$, $\delta_{n}=\frac{\beta_{0}}{n_{0}B_{n,R}}$, and $n_{0}$ is the noise power spectrum density.

Similarly, the channel capacity between the UAV and the EVE as well as the BS can be expressed as follows, respectively
(5)$$C_{EVE}=\sum_{n\in \Omega }B_{n,R}\log_{2}\left(1+\frac{P_{n,R}\delta_{n}}{d_{EVE,R}^{2}}\right)$$
(6)$$C_{BS,R}=W\log_{2}\left(1+\frac{P_{BS}\delta_{BS,R}}{d_{BS,R}^{2}}\right)$$
where *W* is the bandwidth between the R and BS, $\delta_{BS,R}=\frac{\beta_{0}}{\sigma_{BS,R}^{2}}$, and $\sigma_{BS,R}^{2}=n_{0}W$ is the noise power at the UAV.

According to [12], the security capacity is defined as the difference between the legitimate channel capacity and the eavesdropping channel capacity, i.e.,
(7)$$C_{\sec}=\sum_{n\in \Omega }C_{n}-C_{EVE}$$

In this paper, we aim to maximize $C_{\sec}$ by jointly optimizing the UAV’s 3D trajectory, bandwidth, and power allocation with the constraints on information causality [27], LoS communication reliability, and the QoS requirement of mobile users. We formulate the problem of interest as follows
(8a)$$\begin{array}{cc} (P1)\; & \;\max_{q,P,B}\;\;\;C_{\sec} \end{array}$$
(8b)$$\begin{array}{cc} \;s.\;t. & \sum_{n\in \Omega }C_{n}\leq C_{BS,R} \end{array}$$
(8c)$$\begin{array}{cc} \; & C_{EVE}<C_{BS,R} \end{array}$$
(8d)$$\begin{array}{cc} \; & \sin\left(\varphi +\frac{1}{\xi }\ln\frac{\varphi \varepsilon_{\mathrm{LoS}}}{1-\varepsilon_{\mathrm{LoS}}}\right)\cdot d_{n,R}\leq z_{R},\forall n\in \Omega \end{array}$$
(8e)$$\begin{array}{cc} \; & \sin\left(\varphi +\frac{1}{\xi }\ln\frac{\varphi \varepsilon_{\mathrm{LoS}}}{1-\varepsilon_{\mathrm{LoS}}}\right)\cdot d_{BS,R}\leq z_{R} \end{array}$$
(8f)$$\begin{array}{cc} \; & \sum_{n\in \Omega }B_{n,R}\leq B_{\max},\forall n\in \Omega \end{array}$$
(8g)$$\begin{array}{cc} \; & \sum_{n\in \Omega }P_{n,R}\leq P_{\max},\forall n\in \Omega \end{array}$$
(8h)$$\begin{array}{cc} \; & \frac{P_{n,R}\delta_{n}}{d_{n,R}^{2}}\geq \lambda_{\min},\forall n\in \Omega \end{array}$$
where (8b) and (8c) are information causality constraints in the relaying system, i.e., UAV R can only forward the data that has already been received from BS during the relaying process [27]; (8d) and (8e) guarantee the reliability of the LoS communication; and (8h) guarantees the QoS requirement of mobile users. In addition, (8f) and (8g) limit the maximum available bandwidth and maximum transmission power.

## 3. Problem Formulation

Note that the above problem contains nonconvex constraints and nonconvex objective functions, and the UAV location, power allocation, and bandwidth allocation are coupled with each other. To solve this problem, we propose an efficient iterative optimization algorithm, i.e., P1 is decomposed into three subproblems to optimize the UAV’s position, power allocation, and bandwidth allocation, respectively.

### 3.1. 3D Trajectory Optimization

In this section, the trajectory of the UAV is optimized with fixed power and bandwidth allocations. The slack variables $\left\{\vartheta_{n},\vartheta_{EVE},\vartheta \right\}\left(\forall n\in \Omega \right)$ are introduced, and the subproblem can be formulated as
(9a)$$\begin{array}{cc} \;(P1.1)\;\max & \underset{q,\vartheta_{n,},\vartheta_{EVE}}{C_{\sec}}=\sum_{n\in \Omega }B_{n,R}\left(\log_{2}\left(1+\frac{P_{n,R}\delta_{n}}{\vartheta_{n}}\right)-\log_{2}\left(1+\frac{P_{n,R}\delta_{n}}{\vartheta_{EVE}}\right)\right) \end{array}$$
(9b)$$\begin{array}{cc} \;s.\;t.\; & \;\sum_{n\in \Omega }B_{n,R}\log_{2}\left(1+\frac{P_{n,R}\delta_{n}}{\vartheta_{n}}\right)\leq C_{BS,R} \end{array}$$
(9c)$$\begin{array}{cc} \; & \sum_{n\in \Omega }B_{n,R}\log_{2}\left(1+\frac{P_{n,R}\delta_{n}}{\vartheta_{EVE}}\right)\leq C_{BS,R} \end{array}$$
(9d)$$\begin{array}{cc} & \vartheta_{n}\geq {\left(x_{R}-x_{n}\right)}^{2}+{\left(y_{R}-y_{n}\right)}^{2}+z_{R}^{2},\forall n\in \Omega \end{array}$$
(9e)$$\begin{array}{cc} \; & \vartheta_{EVE}\leq {\left(S-x_{R}\right)}^{2}+{\left(D-y_{R}\right)}^{2}+z_{R}^{2}+r_{mk}^{2}-2r_{mk}\vartheta \end{array}$$
(9f)$$\begin{array}{cc} \; & \vartheta \geq ∥\left(S-x_{R},D-y_{R}\right)∥ \end{array}$$
(9g)$$\begin{array}{cc} \; & \left(8d),\;(8e),\;(8h\right) \end{array}$$

It can be seen that P1.1 is equivalent to the trajectory optimization subproblem of P1 when the constraints (9d)–(9f) are satisfied with equality. For any ∀n∈Ω, if (9d) or (9e) is strictly unequal, $C_{\sec}$ will increase as $\vartheta_{n}$ decreases or $\vartheta_{EVE}$ increases. Therefore, the optimal solution of P1.1 is necessarily obtained when (9d)–(9f) are satisfied with equality, i.e., the optimal solution to P1 can be obtained by equivalently solving the P1.1 when power and bandwidth allocation are fixed. However, it is still nonconvex programming due to the nonconvex constraints (9b), (9c) and (9e). To solve this problem, the UAV’s location will be updated iteratively based on the SCA algorithm [28].

Define the coordinate of the UAV in the *m*th iteration of the SCA algorithm as $q^{\left(m\right)}=\left(x_{R}^{\left(m\right)},y_{R}^{\left(m\right)},z_{R}^{\left(m\right)}\right)$, the communication rates between BS-R/n-R are $R_{BS,R}^{\left(m\right)}/R_{n,R}^{\left(m\right)}$, respectively. Let $q^{\left(m\right)}-q^{\left(m-1\right)}=\left(\alpha_{R}^{\left(m\right)},\beta_{R}^{\left(m\right)},\varepsilon_{R}^{\left(m\right)}\right)$. According to [28], we can estimate the lower bounds $R_{BS,R}^{(m),lb}/R_{n,R}^{(m),lb}$ /$d_{EVE,R}^{(m),lb}$ of $R_{BS,R}^{\left(m\right)}/R_{n,R}^{\left(m\right)}$/$d_{EVE,R}^{\left(m\right)}$ by a first-order Taylor expansion, i.e.,
(10a)$$\begin{array}{cc} \; & R_{BS,R}^{(m),lb}=R_{BS,R}^{\left(m-1\right)}-a_{BS,R}^{\left(m-1\right)}\left[{\left(\alpha^{\left(m\right)}\right)}^{2}+{\left(\beta^{\left(m\right)}\right)}^{2}+{\left(\varepsilon^{\left(m\right)}\right)}^{2}\right]-b_{BS,R}^{\left(m-1\right)}\alpha^{\left(m\right)}-c_{BS,R}^{\left(m-1\right)}\beta^{\left(m\right)}-e_{BS,R}^{\left(m-1\right)}\varepsilon^{\left(m\right)} \end{array}$$
(10b)$$\begin{array}{cc} \; & R_{n,R}^{(m),lb}=R_{n,R}^{\left(m-1\right)}-a_{n,R}^{\left(m-1\right)}\left(\vartheta_{n}^{\left(m\right)}-\vartheta_{n}^{\left(m-1\right)}\right)\; \end{array}$$
(10c)$$\begin{array}{cc} \; & d_{EVE,R}^{(m),lb}=d_{EVE,R}^{\left(m-1\right)}+2\left(x_{R}^{\left(m-1\right)}-S\right)\alpha^{\left(m\right)}+2\left(y_{R}^{\left(m-1\right)}-D\right)\beta^{\left(m\right)}+2z_{R}^{\left(m-1\right)}\varepsilon^{\left(m\right)}-2r_{mk}\left(\vartheta^{\left(m\right)}-\vartheta^{\left(m-1\right)}\right) \end{array}$$
where $a_{BS,R}^{\left(m\right)}=\frac{P_{BS}\delta_{BS,R}}{\ln2{\left(d_{R,BS}^{\left(m\right)}\right)}^{2}\left[P_{BS}\delta_{BS,R}+{\left(d_{R,BS}^{\left(m\right)}\right)}^{2}\right]}$, $b_{BS,R}^{\left(m\right)}=2x_{R}^{\left(m\right)}a_{R,BS}^{\left(m\right)}$, $c_{BS,R}^{\left(m\right)}=2y_{R}^{\left(m\right)}a_{R,BS}^{\left(m\right)}$, $e_{BS,R}^{\left(m\right)}=2h_{R}^{\left(m\right)}a_{R,BS}^{\left(m\right)}$, $a_{n,R}^{\left(m\right)}=\frac{P_{n,R}\delta_{n}}{\ln2\left(P_{n,R}\delta_{n}+\vartheta_{n}\right)}$.

Based on the above analysis, the SCA algorithm solves P1.1 at the *m*th iteration by transforming it as follows:
(11a)$$\begin{array}{cc} \;(P1.1.1)\;\;\max_{q^{\left(m\right)},\vartheta_{n}^{\left(m\right)},\vartheta_{EVE}^{\left(m\right)}} & C_{\sec}=\sum_{n\in L}B_{n,R}\left(R_{n,R}^{(m),lb}-\log_{2}\left(1+\frac{P_{n,R}\delta_{n}}{\vartheta_{EVE}^{\left(m\right)}}\right)\right) \end{array}$$
(11b)$$\begin{array}{cc} \;s.\;t. & \;\sum_{n\in \Omega }B_{n,R}\log_{2}\left(1+\frac{P_{n,R}\delta_{n}}{\vartheta_{n}^{\left(m\right)}}\right)\leq WR_{BS,R}^{(m),lb} \end{array}$$
(11c)$$\begin{array}{cc} \; & \sum_{n\in \Omega }B_{n,R}\log_{2}\left(1+\frac{P_{n,R}\delta_{n}}{\vartheta_{EVE}^{\left(m\right)}}\right)\leq WR_{BS,R}^{(m),lb} \end{array}$$
(11d)$$\begin{array}{cc} \; & \vartheta_{n}^{\left(m\right)}\geq {∥x_{n}-x_{R}^{\left(m\right)},y_{n}-y_{R}^{\left(m\right)},z_{R}^{\left(m\right)}∥}^{2},\forall n\in \Omega \end{array}$$
(11e)$$\begin{array}{cc} \; & \vartheta_{EVE}^{\left(m\right)}\leq d_{EVE,R}^{(m),lb} \end{array}$$
(11f)$$\begin{array}{cc} \; & \left(8d),\;(8e),\;(8h),\;(9f\right) \end{array}$$

P1.1.1 is a convex optimization problem, which can be solved by CVX. After obtaining the optimal solution $q_{\mathrm{opt}\;}$ of P1.1.1, it will be used as the input for the next iteration. Finally, the optimization result $q_{\mathrm{opt}\;}$ of the UAV trajectory can be obtained by several iterations. In summary, let $X^{\left(0\right)}=q^{\left(0\right)}=\left(x_{R}^{\left(0\right)},y_{R}^{\left(0\right)},z_{R}^{\left(0\right)}\right)$, $X^{\left(m-1\right)}=q^{\left(m-1\right)}$, $X_{\mathrm{opt}}=q^{\left(m_{\mathrm{opt}}\right)}$, ${\mathrm{OP}}^{X}=P1.1.1$; then, the subproblem of trajectory optimization can be solved by Algorithm 1.
**Algorithm 1** SCA algorithm1:Initialize the location matrix $X^{\left(0\right)}$, m=1, define maximum iteration *M*, tolerance of accuracy E, objective function values $C_{\sec\;}^{\left(0\right)}$2:Find the optimal solution $X^{\left(m\right)}$ to problem ${\mathrm{OP}}^{X}$ by CVX, and calculate $C_{\sec}^{\left(m\right)}$3:**If**$\left|C_{\sec}^{\left(m\right)}-C_{\sec}^{\left(m-1\right)}\right|\leq \xi$**or**m≥M    **Update**
$m_{\mathrm{opt}}=m$4:**Output**$X_{\mathrm{opt}\;}=X^{\left(m_{opt}\right)}$5:**Else**    **Let**
m←m+1**, repeat 2**

### 3.2. Power Allocation Optimization

In this subsection, we focus on the subproblem to find the optimal power allocation, while the UAV’s trajectory and bandwidth are fixed. Obviously, the function $R_{EVEn}=\log_{2}\left(1+\frac{P_{n,R}\delta_{n}}{d_{EVE,R}^{2}}\right)$ is concave, so the optimization problem cannot be solved directly. Similar to P1.1.1, we use the SCA algorithm to solve it. Specifically, in the *m*th iteration, the upper bound $R_{EVEn}^{(m),ub}$ of $R_{EVEn}^{\left(m\right)}$ can be estimated by the first-order Taylor expansion as follows
(12)$$R_{EVEn}^{\left(m\right)}\leq R_{EVEn}^{{\left(m\right)}_{,}ub}=R_{EVEn}^{\left(m-1\right)}+\frac{\delta_{n}/d_{EVE,R}^{2}}{\left(1+P_{n,R}^{\left(m-1\right)}\gamma_{n}/d_{EVE,R}^{2}\right)}\left(P_{n,R}^{\left(m\right)}-P_{n,R}^{\left(m-1\right)}\right)$$

The optimization problem can be rewritten as
(13a)$$\begin{array}{cc} \left(P1.2\right)\;\max_{P,r_{n},\vartheta_{EVEn}} & C_{\sec}=\sum_{n\in \Omega }B_{n,R}^{\left(m\right)}\left(r_{n,R}^{\left(m\right)}-R_{EVEn}^{(m),ub}\right) \end{array}$$
(13b)$$\begin{array}{cc} s.\;t.\;\; & \sum_{n\in \Omega }r_{n}^{\left(m\right)}\leq WR_{BS,R} \end{array}$$
(13c)$$\begin{array}{cc} \;\;\; & \sum_{n\in \Omega }B_{n,R}R_{EVEn}^{(m),ub}\leq WR_{BS,R} \end{array}$$
(13d)$$\begin{array}{cc} \; & \left(8g),\;(8h\right) \end{array}$$

P1.2 is then reduced to a convex programming problem and can be solved by CVX. Since the objective function and constraints contain the log(·) function, CVX processing requires a large number of approximation iterations [29]. In order to solve this problem, we introduce two sets of auxiliary variables $\left\{\rho_{n}\right\},\left\{\chi_{n}\right\},\forall n\in \Omega$, and transform the constraints into LMI constraints or SOC constraints. Then, the optimization problem is expressed as
(14a)$$\begin{array}{cc} (P1.2.1)\;\;\max_{P,\rho_{n},\chi_{n}} & C_{\sec}=\sum_{n\in \Omega }B_{n,R}^{\left(m\right)}\left(\rho_{n}^{\left(m\right)}-R_{EVEn}^{(m),ub}\right) \end{array}$$
(14b)$$\begin{array}{cc} s.t. & \sum_{n\in \Omega }\rho_{n}^{\left(m\right)}\leq WR_{BS,R} \end{array}$$
(14c)$$\begin{array}{cc} \; & \sum_{n\in \Omega }B_{n,R}R_{EVEn}^{(m),ub}\leq WR_{BS,R} \end{array}$$
(14d)$$\begin{array}{cc} \; & \rho_{n}^{\left(m\right)}\leq r_{n}^{lb,(m)},\forall n\in \Omega \end{array}$$
(14e)$$\begin{array}{cc} \; & \frac{1}{\chi_{n}^{\left(m\right)}}\leq \frac{P_{n,R}^{\left(m\right)}\delta_{n}}{d_{n,R}^{2}},\forall n\in \Omega \end{array}$$
(14f)$$\begin{array}{cc} \; & \left(8g),\;(8h\right) \end{array}$$
where $r_{n}^{lb,(m)}$ is the lower bound obtained by performing a Taylor expansion on $\log_{2}\left(1+\frac{1}{\chi_{n}^{\left(m\right)}}\right)$ at $\chi_{n}^{\left(m-1\right)}$, i.e.,
(15)$$r_{n}^{lb,(m)}=\log_{2}\left(1+\frac{1}{{\chi_{n}}^{\left(m-1\right)}}\right)-\frac{1}{\ln2}\frac{1}{{\chi_{n}}^{\left(m-1\right)}\left(1+\chi_{n}^{\left(m-1\right)}\right)}\left({\chi_{n}}^{\left(m\right)}-{\chi_{n}}^{\left(m-1\right)}\right)$$

For the constraint (14e), it can be transformed into an SOC constraint by simple transformation, i.e.,
(16)$$∥\left(P_{n,R}^{\left(m\right)},\chi_{n}^{\left(m\right)},d_{n,R}\sqrt{2/\delta_{n}}\right)∥\leq P_{n,R}^{\left(m\right)}+\chi_{n}^{\left(m\right)}$$

P1.2.1 is a simple second-order cone programming (SOCP) problem, which can also be solved by CVX effectively. In summary, let $X^{\left(0\right)}=P^{\left(0\right)},\;X^{\left(m-1\right)}=P^{\left(m-1\right)},$$X_{\mathrm{opt}\;}=P^{\left(m_{\mathrm{opt}\;}\right)},$${\mathrm{OP}}^{X}=P1.2.1$; then, the subproblem of power allocation optimization can be solved by Algorithm 1.

### 3.3. Bandwidth Allocation Optimization

This section investigates the optimization of bandwidth allocation with a fixed location of UAV and power allocation.

Similar to Section 3.2, in the iterative process using the SCA algorithm, we introduce auxiliary variables $\left\{t_{n}\right\},\left\{\mu_{n}\right\},n\in \Omega$ to transform the constraints into LMI or SOC constraints. Then, the optimization problem is expressed as follows
(17a)$$\begin{array}{cc} \left(P1.3\right)\;\;\max_{B,\left\{t_{n}\right\},\left\{\mu_{n}\right\}} & C_{\sec}=\sum_{n\in \Omega }t_{n}^{\left(m\right)}-\sum_{n\in \Omega }C_{EVEn}^{(m),ub} \end{array}$$
(17b)$$\begin{array}{cc} \;s.\;t. & \sum_{n\in \Omega }t_{n}^{\left(m\right)}\leq WR_{BS,R} \end{array}$$
(17c)$$\begin{array}{cc} \; & \sum_{n\in \Omega }C_{EVEn}^{(m),ub}\leq WR_{BS,R} \end{array}$$
(17d)$$\begin{array}{cc} \; & t_{n}^{\left(m\right)}\leq \mu_{n}^{\left(m\right)}\tau_{n,R}^{(m),lb},\forall n\in \Omega \end{array}$$
(17e)$$\begin{array}{cc} \; & \mu_{n}^{\left(m\right)}\leq B_{n,R}^{\left(m\right)},\forall n\in \Omega \end{array}$$
(17f)$$\begin{array}{cc} \; & \left(8f),\;(8h\right) \end{array}$$
where $C_{EVEn}^{(m),ub}$ is the upper bound obtained by the first-order Taylor expansion of $C_{EVEn}^{\left(m\right)}=\log_{2}\left(1+\frac{P_{n,R}\beta_{0}}{n_{0}B_{n,R}^{\left(m\right)}d_{EVE,R}^{2}}\right)$ at $B_{n,R}^{\left(m-1\right)}$, and $\tau_{n,R}^{(m),lb}$ is the lower bound obtained by the first-order Taylor expansion of $\tau_{n,R}^{\left(m\right)}=\log_{2}\left(1+\frac{P_{n,R}\beta_{0}}{n_{0}B_{n,R}^{\left(m\right)}d_{n,R}^{2}}\right)$ at $B_{n,R}^{\left(m-1\right)}$. $C_{EVEn}^{(m),ub}$ and $\tau_{n,R}^{(m),lb}$ are approximated by the following convex quadratic functions:(18)$$\begin{array}{cc} \; & C_{EVEn}^{(m),ub}=\\ \; & C_{EVEn}^{\left(m-1\right)}+\left(\log_{2}\left(1+\frac{l_{n}}{B_{n,R}^{\left(m-1\right)}}\right)-\frac{l_{n}}{\ln2\left(\tau_{n}+B_{n,R}^{\left(m-1\right)}\right)}\right)\left(B_{n,R}^{\left(m\right)}-B_{n,R}^{\left(m-1\right)}\right)\geq C_{EVEn}^{\left(m\right)},l_{EVEn}=\frac{P_{n,R}\beta_{0}}{n_{0}d_{EVE,R}^{2}} \end{array}$$
(19)$$\begin{array}{cc} \; & \tau_{n,R}^{(m),lb}=\log_{2}\left(1+\frac{l_{n}}{B_{n,R}^{\left(m-1\right)}}\right)-\frac{1}{\ln2}\frac{l_{n}}{B_{n,R}^{\left(m-1\right)}\left(\tau_{n}+B_{n,R}^{\left(m-1\right)}\right)}\left(B_{n,R}^{\left(m\right)}-B_{n,R}^{\left(m-1\right)}\right)\leq r_{n,R}^{\left(m\right)},l_{n}=\frac{P_{n,R}\beta_{0}}{n_{0}d_{n,R}^{2}} \end{array}$$

P1.3 is a simple SOCP problem, which can also be solved by CVX. In summary, let $X^{\left(m-1\right)}=B^{\left(m-1\right)}$, $X_{\mathrm{opt}\;}=B^{\left(m_{\mathrm{op}\;}\right)}$, ${\mathrm{OP}}^{X}=P1.3$; then, the subproblem of bandwidth optimization can be solved by Algorithm 1.

### 3.4. Joint Optimization Algorithm

Combining the analysis of the three subproblems, a joint optimization algorithm for secrecy capacity is proposed, which uses Alternating Iterative Optimization until the objective function converges.

Algorithm Complexity Analysis

(1) P1.1.1 mainly involves $n_{1}=N+5$ variables, i.e., $\alpha_{R}^{\left(m\right)},\beta_{R}^{\left(m\right)},\varepsilon_{R}^{\left(m\right)},{\left\{\vartheta_{n},\vartheta ,\vartheta_{EVE}\right\}}_{n\in \Omega }$;

(2) According to the definitions of LMI constraints and SOC constraints, (8d), (8e), (8h), and (11d) are the SOC constraints of dimension 4 with a total number of 3N+1, and (11e) is an LMI constraint of dimension 5.

(3) Constraint (11b) can be simply expressed in the following form
(20)$$\begin{array}{c} ∥\left(\sqrt{a}\alpha_{R}^{\left(m\right)}+\frac{b}{2\sqrt{a}}\right),\left(\sqrt{a}\beta_{R}^{\left(m\right)}+\frac{c}{2\sqrt{a}}\right),\left(\sqrt{a}\varepsilon_{R}^{\left(m\right)}+\frac{e}{2\sqrt{a}}\right)∥\\ \leq \sqrt{R-\sum_{n\in \Omega }B_{n,R}\log_{2}\left(1+\frac{P_{n,R}\beta_{0}}{\vartheta_{n}^{\left(m\right)}n_{0}B_{n,R}}\right)/W+\frac{b^{2}}{4a}+\frac{c^{2}}{4a}+\frac{e^{2}}{4a}} \end{array}$$

Thus, (11b) is an SOC constraint of dimension 4. Similarly, (11c) is also an SOC constraint of dimension 4. Combining (2) and (3), it is known that P2.1 has a total of 3N+3 SOC constraints of dimension 4. In summary, we can see that Algorithm 1 contains one LMI constraint of dimension 5, 3N+3 SOC constraints of dimension 4, and one SOC constraint of dimension 3; thus, the complexity of the algorithm can be expressed as $O_{1}=n_{1}\sqrt{6N+13}\left[25n_{1}+48N+182+n_{1}^{2}\right]$.

Similarly, the complexity of subproblems 2 and 3 are $O_{2}=n_{2}\sqrt{7N}[3N^{3}+3N^{2}n_{2}+$$2n_{2}N+6N+n_{2}^{2}],n_{2}=3N$, $O_{3}=n_{3}\sqrt{7N}\left[3N^{3}+3n_{3}N^{2}+6Nn_{3}+10N+n_{3}^{2}\right]$, and $n_{3}=3N$, respectively. In summary, the complexity of the joint optimization algorithm is $o=K_{\max}\left(O_{1}+O_{2}+O_{3}\right)$.

Proof of convergence

Combined with [24], it can be seen that each subproblem obtains a lower bound on the original optimization problem, and the objective function is monotonically non-decreasing during the alternating iterations, i.e.
(21)$$\begin{array}{cc} \; & C_{\sec}\left(q^{\left(k\right)},P^{\left(k\right)},B^{\left(k\right)}\right)\\ \; & \leq C_{\sec}\left(q^{\left(k+1\right)},P^{\left(k\right)},B^{\left(k\right)}\right)\\ \; & \leq C_{\sec}\left(q^{\left(k+1\right)},P^{\left(k+1\right)},B^{\left(k\right)}\right)\\ \; & \leq C_{\sec}\left(q^{\left(k+1\right)},P^{\left(k+1\right)},B^{\left(k+l\right)}\right) \end{array}$$

Thus, P1 can be obtained by Algorithm 2 with finite iterations to obtain a convergent solution.
**Algorithm 2** Joint optimization of coordinate and resource allocation algorithm1:Initialize $q^{\left(0\right)}=\left(x_{R}^{\left(0\right)},y_{R}^{\left(0\right)},z_{R}^{\left(0\right)}\right),\;P^{\left(0\right)},\;B^{\left(0\right)}$, $C_{\sec}^{\left(0\right)},k=0$, define maximum iteration $K_{\max}$, tolerance of accuracy ε2:**while**$\left(k\leq K_{\max}\right)\&\left(\Delta >\varepsilon \right)$3:Let $P=P^{\left(k\right)},\;B=B^{\left(k\right)}$, solve problem (1) with Algorithm 1, find optimal coordinate $q^{*}=\left(x_{R}^{*},y_{R}^{*},z_{R}^{*}\right)$4:Let $q^{\left(k+l\right)}=q^{*},\;B=B^{*}$, solve problem (2) with Algorithm 1, find optimal power allocation $P^{*}$5:Let $q^{\left(k+l\right)}=q^{*},P^{\left(k+1\right)}=P^{*}$, solve problem (3) with Algorithm 1, find optimal bandwidth allocation $B^{*}$6:**Update**$C_{\sec}^{\left(k+1\right)},\;\Delta =\left|C_{\sec}^{k+1}-C_{\sec}^{k}\right|/C_{\sec}^{k}$, let k=k+17:**End while**8:**Output**$q^{\left(k\right)},\;P^{\left(k\right)},\;B^{\left(k\right)}$

## 4. Simulation Results

In this section, we assume that the mobile users are randomly distributed in a circle area with the radius of 500 m, and $d_{BS,cen}$ represents the distance between the center of the circle and the BS. Other simulation parameters are given in Table 1.

Figure 2 provides the secrecy capacity versus the number of iterations under different parameters, which can show the convergence characteristics of the proposed joint optimization algorithm. It can be seen that when the number of iterations is larger than 15, the secrecy capacity tends to be stable. Therefore, the proposed optimization algorithm has good convergence under different conditions. In addition, it can be seen that the number of iterations required for the convergence of the algorithm is mainly influenced by the convergence accuracy and the number of ground users. The number of iterations of the algorithm decreases with the value of ε. The number of iterations required for the algorithm to converge also increases with the number of ground users due to more variables being involved.

Considering the mobility of ground users, Figure 3a shows the trajectory of the UAV relay versus the network mobility to clearly describe the relationship between the UAV’s position and mobile users. Figure 3a shows that the center of the mobile user group makes a circular motion, and the number and coordinates of mobile users are randomly generated after each location update. From Figure 3a, the UAV’s height increases gradually with the distance between the mobile users and the BS. The reason is that the flight altitude of the UAV needs to be increased to raise the elevation angle between mobile users and UAV with the increasing communication distance so as to ensure the probability of LoS communication.

Figure 3b shows the projection of the UAV trajectory on the horizontal plane. For ease of elaboration, the projection of the UAV in horizontal plane is denoted as $Q_{UAV}$, and the corresponding center of mobile users is denoted as $Q_{cen}$. It can be seen that $Q_{UAV}$ is always on the right side of $Q_{cen}$, which is influenced by the information causality constraint and the eavesdropper’s position. The offset between $Q_{UAV}$ and $Q_{cen}$ is more pronounced when the ground user is far from the BS and close to the EVE.

To illustrate the effectiveness of the proposed designs, the proposed joint trajectory and communication resource optimization scheme is compared with other three benchmark schemes proposed in [15,24,30].

Scheme 1—The UAV is deployed at the center of mobile users. The height of UAV relay, bandwidth and power allocation are designed to maximize the secrecy communication capacity of this system [30].

Scheme 2—The transmission power and bandwidth are evenly allocated following the principle of fairness [24], and the 3D location of the UAV is optimized.

Scheme 3—With fixed altitude of the UAV relay [15], the 2D trajectory, power and bandwidth allocation are optimized, and the probability of LoS is considered in the simulation process.

From Figure 4, it can be seen that the secrecy capacity obtained by all schemes is decreasing with the distance between the BS and the center of mobile users increasing. This can be easily understood: as the communication distance increases, the signal experiences more fading, and the legitimate channel capacity will decrease. In addition, as the UAV moves far away from the BS, the distance between the UAV and the EVE decreases, causing an increase in eavesdropping channel capacity; then, the secrecy capacity decreases. In scheme 1, the UAV is always located in the mobile users’ center, and the downlink channel quality in the UAV-enabled relay system is better than other schemes at this time. However, this scheme ignores the impact on EVE and BS, resulting in the limitation of legitimate channel capacity by information causality constraint and the increasement of eavesdropping channel capacity. Scheme 2 shows that the power and bandwidth are evenly allocated based on the principle of fairness. For users with relatively poor channel quality, the allocation of the same communication resources cannot achieve the optimal overall performance. In scheme 3, UAV trajectory is optimized at a fixed altitude, which is difficult to ensure the reliability of LoS communication, so the system performance will be worse than the proposed algorithm.

Figure 5 shows the secrecy capacity $C_{\sec}$ versus $B_{\max}$ for different maximum available powers $P_{\max}$. It can be seen that at a certain value of $P_{\max}$, $C_{\sec}$ firstly increases with $B_{\max}$ and then remains constant. This is due to the fact that the relay communication process is limited by the information causality constraint; i.e., the maximum secrecy capacity of the system is constrained by the channel capacity between the BS and the UAV. In other words, increasing power and bandwidth cannot consistently improve the communication performance of the relay system.

Figure 6 shows the secrecy capacity of each mobile user under different schemes. For each mobile user, the algorithm proposed in the paper also has better performance than existing schemes. The reason is that the reliability of LoS communication is considered in the proposed optimization scheme, and we make full use of the location information to achieve dynamic multi-domain optimization. It can be seen that the proposed algorithm is more adaptable to the scenario of multiple mobile users.

## 5. Conclusions

In this paper, we investigate the secrecy capacity optimization problem when the UAV relay serves multiple mobile ground users in the presence of a malicious user eavesdropping on the ground. A joint optimization algorithm for trajectory design and resource allocation is proposed based on the proposed SCA-AIO algorithm. The secrecy capacity of the relay system is improved by jointly optimizing the UAV’s trajectory, power allocation, and bandwidth allocation while ensuring the reliability of LoS communication links, QoS requirements of users, and the information causality constraint. At the end of the paper, the effectiveness of the proposed optimization algorithm is demonstrated by numerical results. In addition, this paper reveals the impacts of different parameters on the system performance and the UAV trajectory trend of UAV relay with mobile ground users. In the text, to ensure the reliability of LoS communication, the UAV’s altitude changes drastically. Considering the limited energy of the UAV, the proposed optimization algorithm will affect the endurance of the UAV relay. In practical applications, it may be necessary to sacrifice some throughput performance to extend the endurance of the UAV. Futhermore, an energy efficiency (EE) maximization UAV trajectory design based on fly–hover will be the next step in our research to compromise the energy consumption and throughput in a UAV-enabled communication system.

## Figures and Tables

**Figure 1 sensors-22-04519-f001:**
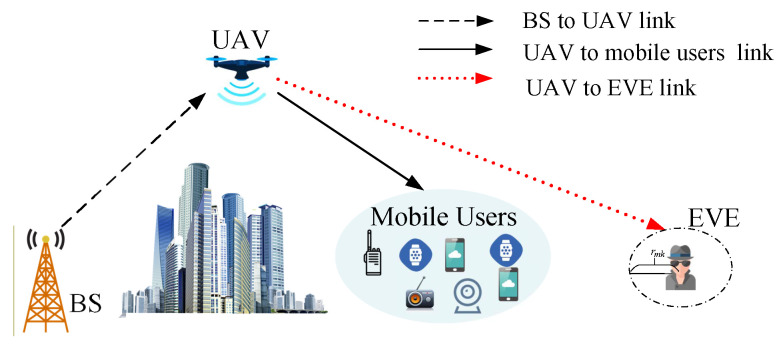
UAV-enabled relaying system model.

**Figure 2 sensors-22-04519-f002:**
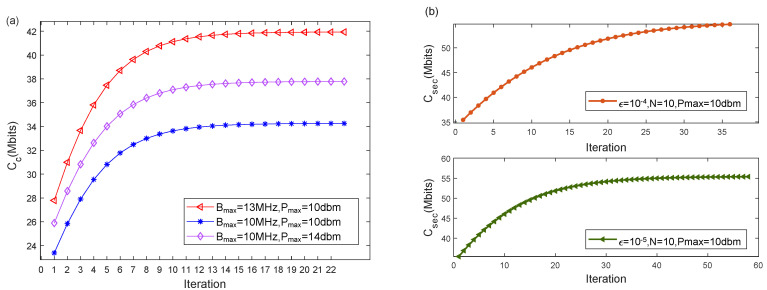
Secrecy communication capacity vs. iterations. (**a**) N = 5, $\varepsilon =10^{-4}$. (**b**) N = 10, $B_{\max}$ = 10 MHz.

**Figure 3 sensors-22-04519-f003:**
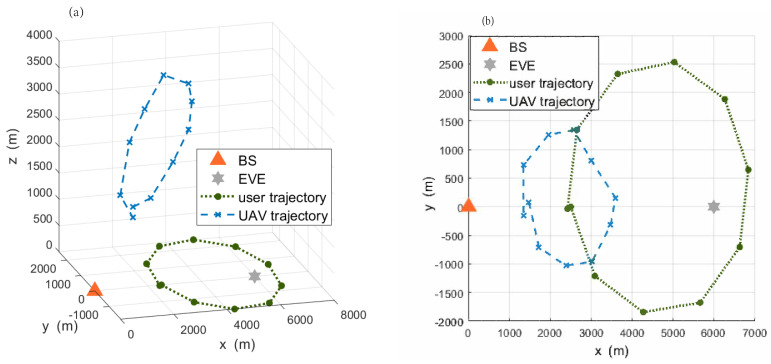
The trajectory of UAV relay versus the network mobility. (**a**) The 3D trajectory of UAV relay. (**b**) Plane projection of UAV trajectory.

**Figure 4 sensors-22-04519-f004:**
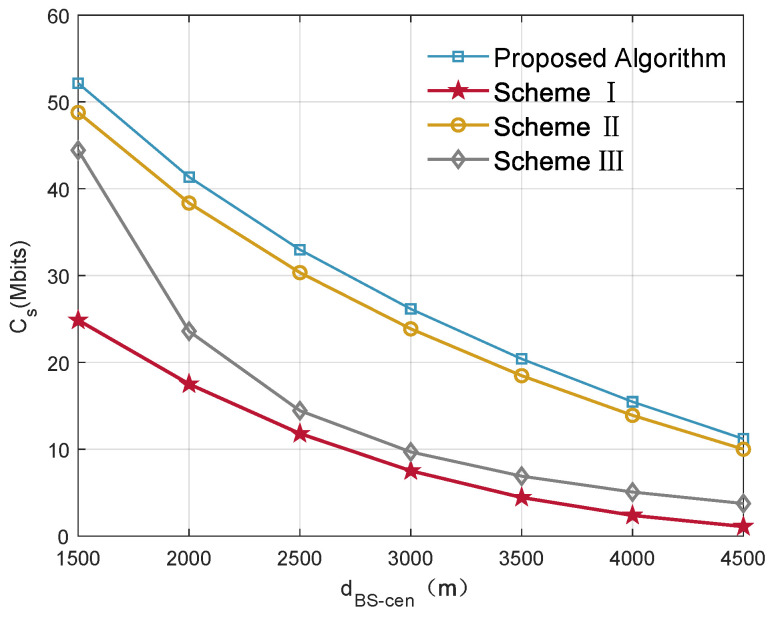
System performance comparisons with distance between the BS and the center of mobile users.

**Figure 5 sensors-22-04519-f005:**
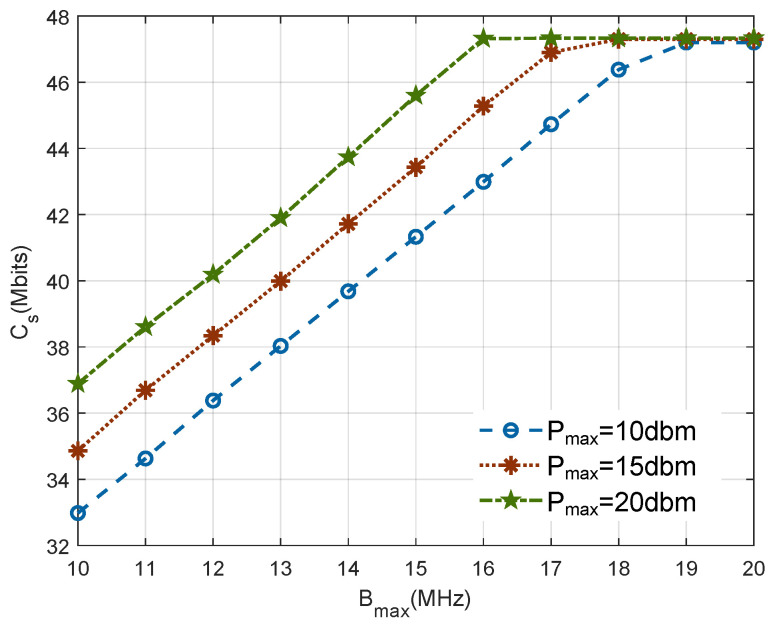
Secrecy communication capacity v.s $B_{\max}$ with different values of $P_{\max}$.

**Figure 6 sensors-22-04519-f006:**
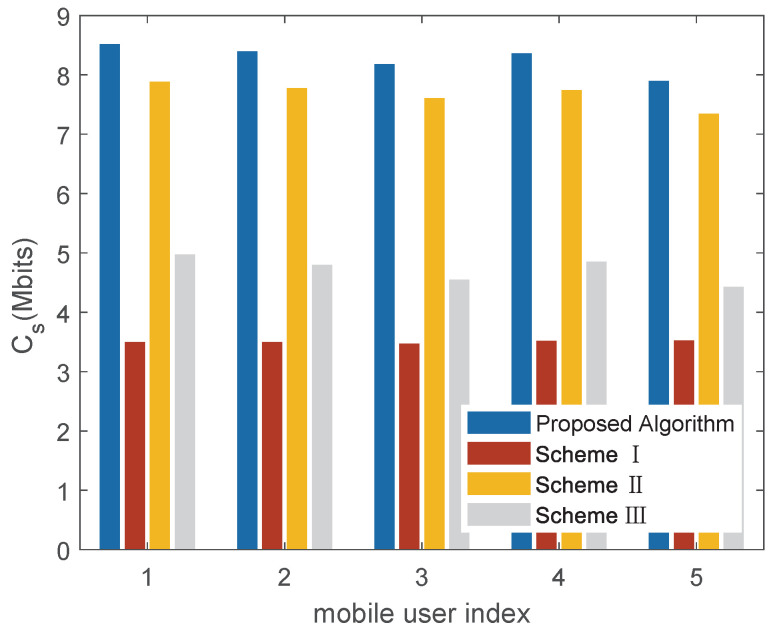
Secrecy capacity of each mobile user under different schemes ($d_{BS,cen}=2000\;m$).

**Table 1 sensors-22-04519-t001:** Simulation parameters.

Parameter	Value	Parameter	Value
*W*	10 MHz	$P_{BS}^{\max}$	30 dbm
$\lambda_{\min}$	1 db	$n_{0}$	−169 dbm/Hz
$\beta_{0}$	−22 dbm	N	5
$P_{\max}$	10 dbm	$q_{\mathrm{EVE}\;}$	(6000, 0, 0)
$\varepsilon_{\mathrm{LoS}}$	0.95	ψ	11.95
$d_{BS.cen}$	2500 m		

## Data Availability

The data presented in this study are available on request from the corresponding author.

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
