# Peer review of "Secrecy Capacity Maximization of UAV-Enabled Relaying Systems with 3D Trajectory Design and Resource Allocation"

_sensors, 2022, doi:10.3390/s22124519_

Round 1

Reviewer 1 Report

The proposed work is interesting, however I have some concerns regarding the benefits of the proposed 3D Trajectory Design and Resource Allocation in real applications. A relevant topic is power consumption of the UAV whic it is not considered. The contribution in theory is interesting however a depper description on the power consumption of the UAV must be address to demonstrate the limitations of the proposed work in real time applications. 

With this consideration I am confident that the work will be more significant for the community and ready for publication.

Reviewer 2 Report

1.      What are the main questions that remained to be addressed by the authors of the article? 

There are no MAIN questions opened because the proposed work is, in my opinion, complete and solid. For a better understanding of the material, I would suggest the Authors address these MINOR points:

A.      Fig. 2: What are the parameters that influence the number of iterations at which the convergence of the algorithm becomes to show up? The Authors are invited to add a few comments in the text.

B.      Fig. 3a and 3b: both figures are too small. The labels and the axis are difficult to be read. The Authors are invited to make them readable.

Reviewer 3 Report

In this manuscript, an optimization algorithm has been proposed to yield jointly the optimal trajectory and resource allocation (in terms of bandwidth and power) for a UAV-assisted communication system. The objective is to also maximize the secrecy capacity to combat the eavesdroppers in the open wireless medium. Overall, the details of the optimization problem along with the steps for solving the problem had been presented quite clearly in both Sections 2 and 3. The numerical examples in Section 4 had also demonstrated quite well the added benefits of proposed joint allocation algorithm when compared with the performance of other and similar allocation schemes. The results presented in this manuscript would be of significant interest to those working in the same research field.

Author Response

Thank you for your positive feedback on this article. We will continue to conduct in-depth research on this basis.